# Morphological and Phylogenetic Analyses Reveal Three Novel Species of *Sanguinoderma* (*Ganodermataceae, Basidiomycota*) from Yunnan Province, China

**DOI:** 10.3390/jof10080589

**Published:** 2024-08-19

**Authors:** Kai-Yang Niu, Jun He, Song-Ming Tang, Xi-Jun Su, Zong-Long Luo

**Affiliations:** 1College of Agriculture and Biological Science, Dali University, Dali 671003, China; niukaiyang@126.com (K.-Y.N.); tang202205@gmail.com (S.-M.T.); suxijundali@163.com (X.-J.S.); 2College of Biotechnology and Engineering, West Yunnan University, Lincang 677000, China; 3Cangshan Forest Ecosystem Observation and Research Station of Yunnan Province, Dali University, Dali 671003, China

**Keywords:** *Sanguinoderma*, morphological study, new taxa, phylogenetic analysis

## Abstract

*Sanguinoderma* (*Ganodermataceae*) is recognized as a valuable medicinal resource in Taiwan, China. Additionally, it serves as a traditional folk medicine for treating neurotic epilepsy in Malaysia. This study involved the collection of six specimens of *Sanguinoderma* from Yunnan Province, China. Employing multigene phylogenetic analysis of DNA sequences, including internal transcribed spacer (ITS), nuclear large subunit (LSU), RNA polymerase II second largest subunit (*rpb*2), translation elongation factor 1-alpha (*tef*1-α), mitochondrial small subunit (mtSSU), nuclear small subunit (nSSU) and morphological examinations, three new species, viz. *Sanguinoderma concentricum*, *S. dehongense* and *S. ovisporum*, are introduced. *Sanguinoderma concentricum* is characterized by a central stipe basidiomata, an orbicular to suborbicular pileus, a grayish-yellow surface with alternating concentric zones and wavy margin-like petals and regular pileipellis cells (4–8 × 17–28 μm). *Sanguinoderma dehongense* is characterized by a long stipe and flabelliform basidiomata, a dark-grayish yellow-to-dark-yellow pileus surface, irregular pileipellis cells and wavy margin and ellipsoid basidia (8–11 × 9–13 μm). *Sanguinoderma ovisporum* is characterized by a reniform basidiomata, a heterogeneous context and ovoid basidiospores (7.5–8.6 × 5.5–7.2 µm). A detailed description and illustrations of these new species are provided, as well as a morphological comparison with similar taxa.

## 1. Introduction

*Ganodermataceae* is a genera-rich and highly diverse group of macro-fungi, widely distributed in tropical and subtropical regions around the world. Up to date, fourteen genera are accepted in *Ganodermataceae*. Previous studies have confirmed that species within the genera *Ganoderma* P. Karst and *Sanguinoderma* Y.F. Sun, D.H. Costa and B.K. Cui possess significant medicinal value, including anti-cancer, anti-bacterial, anti-aging, immune-enhancing and lipid-lowering effects [1,2,3,4,5,6,7].

*Sanguinoderma* (*Ganodermataceae*, *Polyporales*) was introduced to accommodate the species in which the fresh pore surface changes to blood red when bruised, with *S. rude* Y.F. Sun, D.H. Costa and B.K. Cui as type species. *Sanguinoderma* is characterized by an annual basidiomata, central or lateral stipitate with an almost sessile, corky-to-woody hard texture. The pileus is single, suborbicular to flatly reniform and glabrous to tomentose, with concentric zonation or furrowing and radial rugosity; the pore surface changes to blood red when bruised [8]. There are 18 epithets of *Sanguinoderma* in the Species Fungorum (http://www.speciesfungorum.org/; accessed date: 13 July 2024). Before the establishment of this genus, the species had consistently been classified within the *Amauroderma* Murrill. In 1905, Murrill [9] established *Amauroderma*, classifying it within the *Ganodermataceae*. Costa-Rezende et al. [10,11] suggested that *Amauroderma* was polyphyletic based on phylogenetic analysis, analyzing the ITS, LSU, *rpb*1, and *tef*1-α sequences of *Ganodermataceae*, and described two new genera, *Furtadoa* Costa-Rez., Drechsler-Santos and Reck and *Foraminispora* Costa-Rez., Drechsler-Santos and Robledo from *Amauroderma*. The rest of the species previously considered *Amauroderma* to be clustered in unrelated clades, one of them considered to be *Amauroderma* s.str., and the other as the ‘*Amauroderma rude*’ clade, but without morphological differences between the two clades. Until 2020, Sun et al. [8] clarified the taxonomy and phylogeny of *Amauroderma* s.lat, when *Sanguinoderma* was established as a new genus.

*Sanguinoderma* is the largest group of species aside from *Ganoderma* P. Karst and *Amauroderma* in *Ganodermataceae*. *Sanguinoderma rugosum* (Blume and T. Nees) Y.F. Sun, D.H. Costa and B.K. Cui is regarded as a precious traditional medicine with anti-cancer properties in Taiwan, China [12], and the indigenous peoples in Peninsular Malaysia used *S. rugosum* to treat epilepsy [13]. Most *Sanguinoderma* species are soil-inhabiting, distributed in tropical and subtropical regions of Africa, Asia, North America, Oceania and South America [8,14].

In China, species of *Sanguinoderma* have been reported in 13 provinces (Chongqing, Fujian, Guangdong, Guangxi, Guizhou, Hainan, Hubei, Hunan, Jiangxi, Sichuan, Taiwan, Yunnan and Zhejiang), and Yunnan Province has the most abundant diversity of species distribution, with a total of seven species [*S. elmerianum* (Murrill) Y.F. Sun and B.K., *S. guangdongense* B.K. Cui and Y.F. Sun, *S. laceratum* Y.F. Sun and B.K. Cui, *S. leucomarginatum* B.K. Cui and Y.F. Sun, *S. longistipitum* B.K. Cui and Y.F. Sun, *S. preussii* (Henn.) B.K. Cui and Y.F. Sun and *S. rugosum*.]; *S. laceratum* and *S. leucomarginatum* are reported exclusively in Yunnan Province [10,11]. In this study, six *Sanguinoderma*-like samples collected from Yunnan Province were identified as three new species based on phylogenetic analysis and morphological study, namely *S. concentricum, S. dehongense* and *S. ovisporum*.

## 2. Materials and Methods

### 2.1. Specimen Collection

Six *Sanguinoderma*-like specimens were collected during July and August of 2023 in the Yunnan Province, China. The collection and documentation process for *Sanguinoderma* specimens entail capturing photographs and recording essential information, such as habitat, altitude, collection time and location. Detailed descriptions of morphological features were also documented. In the benchtop (AIRTECH SW-CJ-2F), a 1–2 mm^2^ of context comprised inoculated tissue on a PDA (peel potatoes: 200 g/L, glucose: 20 g/L, agar: 20 g/L and distilled water: 1000 mL) agar medium to obtain a pure culture. The specimens are then dried in an oven at 40~50 °C and placed with an appropriate amount of silica gel in self-sealing bags to prevent moisture regain. Specimens were deposited in the herbarium of Cryptogams, Kunming Institute of Botany, Academia Sinica (KUN-HKAS).

### 2.2. Morphological Observations

Macro-morphological studies were conducted following the protocols provided by Sun et al. [14]; the color was compared to the standard colors on the colorhexa website (https://www.colorhexa.com (accessed on 10 July 2024)). Micro-morphological structures were obtained from the dried specimens and then photographed by using a Nikon ECLIPSE Ni-U compound microscope fitted with a Nikon DS-Ri2 digital camera. Microscopic observations and color reactions were, respectively, made from slide preparations using 5% potassium hydroxide (KOH), Melzer’s reagent, and Cotton Blue. Measurements were made using the Image Frame work v.0.9.7. At least 20 basidiospores were measured in each specimen, 5% of measurements were excluded from each end of the range and extreme values were provided in parentheses [15]. The following abbreviations were used: IKI = Melzer’s reagent; IKI− = neither amyloid nor dextrinoid; CB = Cotton Blue; CB+ = cyanophilous; L = mean spore length (arithmetic average of all spores); W = mean spore width (arithmetic average of all spores); Q = L/W ratio and (a) = number of spores measured and (b) = specimens number [8]. Ultrastructures of basidiospores were observed via Scanning Electron Microscopy (SEM) at Yunnan Academy of Agricultural Sciences, China.

### 2.3. Growth Rate of Mycelium on Culture Media

A 7 mm diameter culture was taken from the edge of the PDA medium and inoculated onto new PDA, CMA (cornstarch 30 g, agar 20 g and distilled water 1000 mL) and LB (beef extract 3 g/L, peptone 10 g/L, NaCl 5 g/L, agar 20 g/L and distilled water 1000 mL) media. The cultures were incubated at a constant temperature of 24 °C, avoiding light. The growth diameter was measured every 24 h. Each experiment was repeated five times, and the growth rate of the pure culture was recorded.

### 2.4. DNA Extraction, PCR Amplification and Sequencing

Genomic DNA was extracted from dried specimens using Ezup Column Fungi Genomic DNA Purification Kit (Sangon Biotech Limited Company, Kunming, Yunnan, China) based on the manufacturer’s protocol. The internal transcribed spacer (ITS) regions were amplified with primer pairs ITS5 and ITS4 [16], LR0R and LR5 for the large subunit of nuclear ribosomal RNA gene (LSU) [17] and primer pairs fRPB2-5F and fRPB2-7CR [18] were used to amplify the second subunit of RNA polymerase II (*rpb*2). The translation elongation factor 1-α gene (*tef*1-α) was amplified with primer pairs EF1-983F and EF1-1567R [19]. The small subunit mitochondrial rRNA gene (mtSSU) was amplified with primer pairs MS1 and MS2 [16], and the small subunit nuclear ribosomal RNA gene (nSSU) was amplified with primer pairs PNS1 and NS41 [16].

The PCR volume contained 1 µL of each primer, 1 µL of extracted DNA, 9.5 µL of ddH_2_O and 12.5 µL of 2× EasyTaq PCR SuperMix (Sangon Biotechnology Co., Kunming, China). PCR cycling schedules for four-gene regions of ITS, LSU, mtSSU and nSSU were based on Sun et al. [20]. The PCR cycle of *tef*1-α was as follows: initial denaturation at 94 °C for 5 min, denaturation at 94 °C for 30 s, denaturation at 55 °C for 30 s, denaturation at 72 °C for 50 s and extension at 72 °C for 10 min; 35 cycles were repeated. The PCR cycle of *rpb*2 was as follows: initial denaturation at 95 °C for 5 min, 95 °C for 1 min, 51 °C for 2 min, 72 °C for 1.5 min for 35 cycles and, finally, extension for 10 min at 72 °C. The PCR amplicons were sent to Sangon Biotech (Kunming, Yunnan, China) for Sanger sequencing. Raw DNA sequences were assembled and edited in Sequencher v.4.1.4, and the assembled DNA sequences were deposited in GenBank (Table 1).

### 2.5. Sequence Alignment and Phylogenetic Analysis

The ITS, LSU, *rpb*2, *tef*1-α, mtSSU and nSSU sequences used in this study were combined into a dataset. *Magoderna subresinosum* was used as the outgroup [20], which is the sister clade of *Sanguinoderma*. Sequences were aligned using the online version of MAFFT v.7 (https://mafft.cbrc.jp/alignment/server/ (accessed on 6 March 2024)) [24] and were manually adjusted in BioEdit v.7.1.3 [25]. Ambiguous aligned regions were excluded from the analyses, and gaps were treated as missing data. The phylogeny website tool “ALTER” [26] was used to convert the Fasta file to Phylip format for RAxML analysis, Aliview and PAUP v.4.0b 10 were used to convert the Fasta alignment file to a Nexus file for Bayesian analysis [27].

### 2.6. Phylogenetic Inference

Based on the combined dataset, the maximum likelihood (ML) analysis was conducted in RAxML-HPC2 v.8.2.3 [28] and implemented on the CIPRES portal (https://www.phylo.org/portal2/login.action (accessed on 6 March 2024)) [29] with the GTR + G model for each gene and 1000 rapid bootstrap (BS) replicates. Since no supported conflict (BS ≥ 75%) was detected among the topologies, the six single-gene alignments were concatenated using Sequence Matrix [30].

Bayesian analysis was performed in MrBayes 3.2 [31] and the best-fit models of sequences’ evolution were estimated with MrModeltest 2.3 [32,33,34]; the selected models were HKY+G for ITS, GTR+I for LSU and nSS, GTR+I+G for mtSSU and *rpb*2 and HKY+I+G for *tef*1-α. The Markov Chain Monte Carlo (MCMC) sampling approach was used to calculate posterior probabilities (PP) [35]. Bayesian analysis of six simultaneous Markov chains was run for 10,000,000 generations, and trees were sampled every 1000 generations. The first 5000 trees, representing the burn-in phase of the analyses, were discarded, while the remaining 1500 trees were used for calculating posterior probabilities in the majority rule consensus tree (the critical value for the topological convergence diagnostic is 0.01).

Bootstrap support values in maximum likelihood (ML) equal to or greater than 75% and Bayesian posterior probabilities (PP) equal to or greater than 0.95 are provided above the nodes. All trees were viewed in FigTree v. 1.4.0 (http://tree.bio.ed.ac.uk/software/figtree/ (accessed on 12 March 2024)) and were edited using Adobe Illustrator CS5 (Adobe Systems Inc., San Jose, CA, USA) [15]. Sequences derived from this study were deposited in GenBank. The final sequence alignments and the phylogenetic trees are available at Figshare (https://figshare.com/ DIO: 10.6084/m9.figshare.26232650 (accessed on 20 July 2024)).

## 3. Results

### 3.1. Phylogenetic Analyses

In this study, 290 sequences of ITS, LSU, *rpb*2, *tef*1-α, mtSSU and nSSU were used to construct phylogenetic trees of *Sanguinoderma*, including 52 ITS, 51 LSU, 37 *rpb*2, 48 *tef*1-α, 51 mtSSU and 51 nSSU. The sequences were obtained from 53 specimens representing 21 taxa in *Sanguinoderma*. The combined six-gene sequence dataset had an aligned length of 5012 total characters including gaps (ITS: 1–542; LSU: 543–1863; mtSSU: 1864–2371; nSSU: 2372–3454; *rpb*2: 3455–4491 and *tef*1-α: 4492–5011).

The tree topologies of the maximum likelihood analysis and the Bayesian analysis were similar. The RAxML analysis of the combined dataset yielded the best scoring tree with a final maximum likelihood value of −13,811.532268. The matrix had 664 distinct alignment patterns, with 20.32% undetermined characters or gaps. ML and BI analyses generated nearly identical tree topologies with minimal variations in statistical support values. Thus, only a ML tree is shown (Figure 1).

In the phylogenetic analysis, six specimens collected from Yunnan, China formed three monophyletic clades, *S. concentricum* (100% ML/1.00 PP), *S. dehongense* (100% ML/1.00 PP) and *S. ovisporum* (99% ML/0.99 PP). *Sanguinoderma ovisporum* and *S. laceratum* (Cui 8155) form sister clades with a good support value (82% ML/0.99 PP, Figure 1). A comparison of the LSU nucleotide bases between *S. ovisporum* (HKAS 135638) and *S. laceratum* (Cui 8155) revealed differences of 21 bps (21/915, including one gap). A comparison of the ITS and *tef*1-α nucleotide bases between *S. dehongense* (HKAS 135636) and *S. elmerianum* (Dai 20634) revealed differences of 12 bps (542/12, no gaps) and 15 bps (15/510, no gaps), respectively. A comparison of the LSU nucleotide bases between *S. concentricum* (HKAS 135640) and *S. laceratum* (Cui 8155) revealed differences of 19 bps (19/896, no gaps).

**Figure 1 jof-10-00589-f001:**
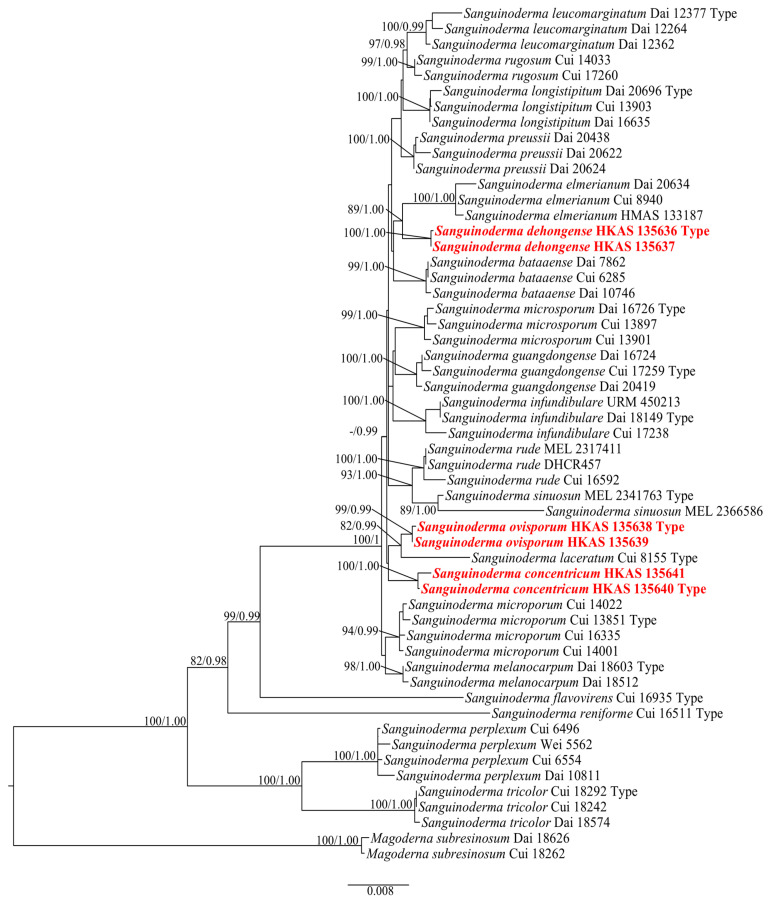
Tree results from the maximum likelihood (ML) phylogenetic tree of *Sanguinoderma* based on the dataset of ITS + LSU + *rpb*2 + *tef*1-α + mtSSU + nSSU. Branches are labeled with maximum likelihood bootstrap values equal to or higher than 75% and Bayesian posterior probability values equal to or higher than 0.95. New species are indicated in bold red.

### 3.2. Taxonomy

***Sanguinoderma concentricum*** K.Y. Niu, J. He and Z.L. Luo sp. nov.


**Fungal Names number: FN 572020**


Figure 2 and Figure 11a,b

**Diagnosis.** *Sanguinoderma concentricum* differs from other species by its central stipe basidiomata, with concentrically zonate pileus surface, wavy margin lacerated like petals and broadly clavate basidia (16–18 × 18–21 μm).

**Etymology.** The epithet ‘*concentricum*’ refers to the pileus surface with obvious concentric zones.

**Holotype.** China, Yunnan Province, Dehong Prefecture, on the ground with humus, 1245 m, Kai-Yang Niu, 7 August 2023, HKAS 135640.

**Description. Basidiomata** is annual, centrally stipitate and occasionally imbricate; coriaceous to corky. **Pileus** orbicular to suborbicular, up to 9.7 cm diameter and 5 mm thick; **surface** is dark and moderately orange (#9d6c4b), dull and tomentose, with alternating concentric zones that are grayish yellow (#d3d2bd) to dark-grayish lime green (#5a605a); dense and radial fine wrinkles; **margin** is moderately orange (#c29144), obtuse, lacerated-like petals, acute, wavy and obviously incurved when dry. **Context** is up to 1 mm thick, homogeneous, light-grayish orange (#f4e8d0) and soft and corky without black melanoid lines. **Tubes** are up to 4 mm long, mostly desaturated dark orange (#b19570), hard and corky and unstratified. **Pores** are 3–4 per mm, circular to angular, grayish yellow (#c8c9c3) when fresh, changing to brown, dark and moderately red (#753a37) when bruised, then quickly darkening, without discoloration and dissepiments entire when dry. **Stipe** is up to 5.1 cm long and 8 mm in diameter, central, cylindrical, hollow, slightly curved, dark-grayish orange (#9c8c7c) and fibrous to woody.

**Hyphal system trimitic**, with generative hyphae 3–6 μm in diameter, hyaline, thin walled and with clamp connections; skeletal hyphae are 5–9 μm in diameter, hyaline, thick walled with a wide-to-narrow lumen, flexuous and arboriform; binding hyphae are 1–3 μm in diameter, hyaline, flexuous and branched. All hyphae are IKI− and CB; tissue darkening in KOH. **Pileipellis is** a regular palisade, apical cells are 4–8 × 17–28 μm, short clavate and yellowish brown. **Basidiospores** are broadly ellipsoid, pale gray, IKI− and CB+ with double and slightly thin walls; exospore wall is smooth; endospore wall features conspicuous pillars, (9.4) 9.6–10.7 (10.9) × (8.1) 8.2–9.2 (9.6), L = 10.0 μm, W = 8.7 μm and Q = 1.15 (40/2); under SEM, exospore wall reticulates (Figure 11a,b). **Basidia**: broadly clavate, hyaline, thin walled and 16–18 × 18–21 μm. **Basidioles** similar to basidia; 18–20 × 23–25 μm.

**Culture features.** Circular, slight smell of corruption, initially white to grayish white, gradually turns grayish black, changing to blood red when bruised, color of the agar does not change with the growth of the culture; generative hyphae feature multiple branches, with irregularly thickened walls or with scattered thick-walled, refractive areas on walls; texture is sub felty and farinaceous (Figure 3). On PDA, at 24 °C, growth is fast, reaching 74.8 mm in 6 days; culture hyphae are sparse. On LB, at 24 °C, growth is slow, reaching 37 mm at 6 days; culture hyphae are dense. On CMA, at 24 °C, growth is extremely slow, reaching 1.5 mm in 6 days; culture hyphae are particularly sparse (Figure 4).

**Additional specimens examined.** China, Yunnan Province, Dehong Prefecture, on the ground with humus, 1539 m, Kai-Yang Niu, 20 August 2023, HKAS 135641.

**Notes:** *Sanguinoderma infundibulare* described from subtropical areas of China is morphologically similar to *S. concentricum* by having the centrally to laterally stipitate basidiomata with a funnel-shape to orbicular pileus. However, *S. infundibulare* differs by the smaller pores (4–6 per mm), pale wood-brown context with dark melanoid lines and larger basidiospore (10–12.2 × 8.5–10.6 μm) [20]. *Sanguinoderma preussii* is another species that also has an orbicular pileus in *Sanguinoderma*, but it differs from *S. concentricum* by the grayish-brown pileus surface, smaller pores (6–7 per mm) and buff yellow context [10]. Moreover, *S. concentricum* and *S. preussii* were supported as two distinct lineages in the phylogenetic tree (Figure 1). In culture, *S. concentricum* exhibited a more uniform mycelial density compared to *S. ovisporum* and *S. dehongense*, with neat colony edges (Figure 3, Figure 6 and Figure 9). On PDA and LB media, the mycelial growth rate of *S. concentricum* was higher than that of *S. ovisporum* and *S. dehongense* (Figure 4, Figure 7 and Figure 10).

**Figure 2 jof-10-00589-f002:**
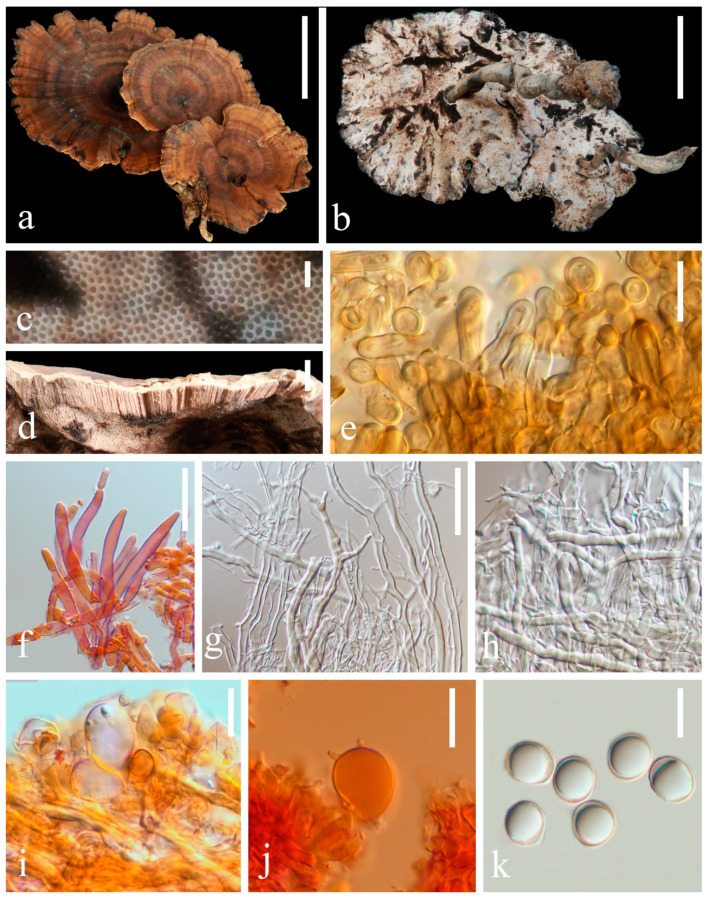
Basidiomata and microscopic structures of *Sanguinoderma concentricum* (HKAS 135640, holotype). (**a**,**b**) Basidiomata. (**c**) Pores. (**d**) Pileus cross-section. (**e**) Apical cells from pileipellis. (**f**) Terminal generative hyphae from tubes. (**g**) Binding and skeletal hyphae from context. (**h**) Skeletal hyphae from context. (**i**) Basidioles. (**j**) Basidia. (**k**) Basidiospores. Scale bars: a, b = 5 cm; c = 1 mm; d = 5 mm; e, i, j = 5 μm; f, h, k = 10 μm and g = 15 μm.

**Figure 3 jof-10-00589-f003:**
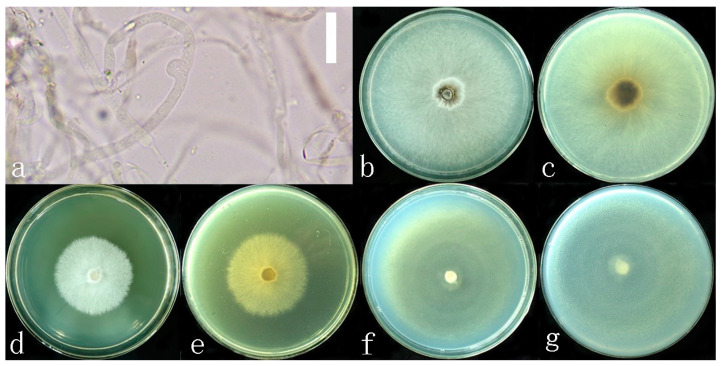
*Sanguinoderma concentricum* culture characters. (**a**) Clamped generative hyphae; (**b**,**c**) colony on PDA; (**b**) obverse; (**c**) reverse; (**d**,**e**) colony LB, (**d**) obverse; (**e**) reverse; (**f**,**g**) colony on CMA; (**f**) obverse; (**g**) reverse. Scale bars: a = 20 μm.

**Figure 4 jof-10-00589-f004:**
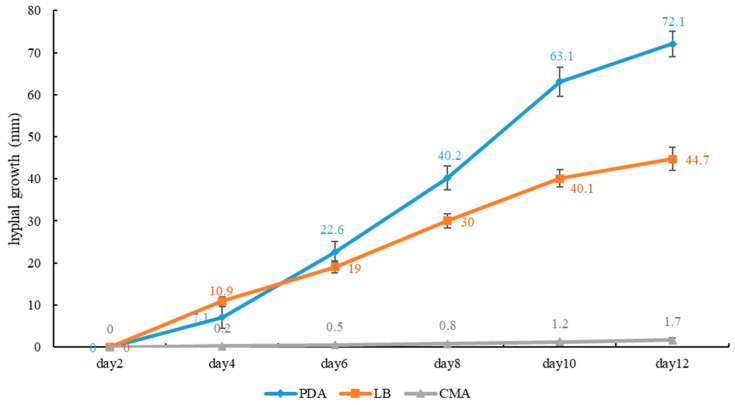
Average daily hyphal growth of *Sanguinoderma concentricum* on different culture media.

***Sanguinoderma dehongense*** K.Y. Niu, J. He and Z.L. Luo sp. nov. 


**Fungal Names number: FN 572019**


Figure 5 and Figure 11c,d

**Diagnosis.** *Sanguinoderma dehongense* differs from other species in its pileus surface being dark grayish yellow to dark yellow with concentrically zonate, long stipe and irregular pileipellis cells and wavy margin.

**Etymology.** The epithet ‘*dehongense*’ refers to the Dehong Prefecture of Yunnan Province, where the holotype was collected.

**Holotype.** China, Yunnan Province, Dehong Prefecture, on the ground covered with humus, 1339 m, Kai-Yang Niu, 7 August 2023, HKAS 135636.

**Description. Basidiomata** is annual, laterally stipitate and is hard and corky to woody and hard. **Pileus** is single, flabelliform to reniform, up to 7 cm in diameter and 5 mm thick; **surface** is grayish yellow (#d1d0be), dull and glabrous with alternating concentric zones that are dark-grayish yellow (#46473f) to dark yellow (#161614); dense and radial fine wrinkles; **margin** is very dark gray (#262626), obtuse, entire, strongly wavy and incurved when dry. **Context** is up to 2 mm thick, homogeneous, slightly desaturated orange (#b59a6b) when dry and soft and corky without black melanoid lines. **Tubes** are up to 3 mm long, dark and moderately orange (#7b5f3d) when dry, hard and woody and unstratified. **Pores** are 3–4 per mm, subregular pentagonal to oblong, very light gray (#fafafa) when fresh, changing to mostly desaturated dark red (#95746c) when bruised, then quickly darkening; mostly desaturated dark orange (#b0926e) and dissepiments entire when dry. **Stipe** is up to 21 cm long and 6 mm diameter, cylindrical, hollow, slightly curved, swollen at base, dark-grayish orange (#a8a190) to very dark-grayish yellow (#33322e) and fibrous to woody.

**Hyphal system trimitic**, with generative hyphae 2–5 μm in diameter, hyaline and thin walled with clamp connections; skeletal hyphae are 4–8 μm in diameter, pale grey to pale yellow and thick walled with a wide to narrow lumen or subsolid, straight and little branched; binding hyphae are 2–5 μm in diameter, pale grey, thick walled, branched and flexuous; all hyphae IKI−, CB+; tissue darkening in KOH. **Pileipellis** is irregular palisade; apical cells are 5–9 × 11–21 μm, tightly packed, with narrow lumen, tightly packed together, thick walled and pale yellowish brown, forming an irregular palisade. **Basidiospores** are subglobose to broadly ellipsoid, pale grey, IKI− and CB, with double and thick walls; exospore wall is smooth; endospore wall features faint pillars, (9.1) 9.3–10.8 (11.0) × (8.1) 8.4–9.8 (9.9) μm, L = 10.1 μm, W = 9.1 μm and Q = 1.11 (40/2). Under SEM, exospore wall is reticulate (Figure 11c,d). **Basidia** is ellipsoid, hyaline, thin walled and 8–11 × 9–13 μm. **Basidioles** are obovoid, hyaline, thin walled and 12–14 × 15–19 μm.

**Culture feature.** Circular, edge irregularity, slight smell of corruption and initially white to grayish white; it gradually turns grayish and eventually ages to black; color changes to blood red when bruised and color does not change with agar-induced growth of the culture; generative hyphae feature multiple branches, with irregularly thickened walls or with scattered thick-walled, refractive areas on the walls; texture is sub felty and farinaceous (Figure 6). On PDA at 24 °C, growth is fast, reaching 72.1 mm in 12 days; culture hyphae are dense. On LB at 24 °C, growth is slow, reaching 44.7 mm in 12 days; culture hyphae are dense. On CMA at 24 °C, growth is extremely slow, reaching 1.7 mm at 12 days; culture hyphae are particularly sparse (Figure 7).

**Additional specimen examined.** China, Yunnan Province, Dehong Prefecture, on the ground covered with humus, 1545 m, Kai-Yang Niu, 7 August 2023, HKAS 135637.

**Notes:** Morphologically, *Sanguinoderma longistipitum* has a similar distribution with *S. dehongense* which can be collected from Yunnan Province, but the former can be distinguished by a sub-orbicular to flabelliform pileus, a smaller basidiomata, a pore density of 6–8 per mm and finger-shaped pileipellis cells with multiple obvious septa [20]. *Sanguinoderma elmerianum* also has lateral stipitate basidiomata and is swollen at the base of the stipe, but it differs from *S. dehongense* by the flat reniform pileus and short stipe (9 cm long) [8]. Based on phylogenetic analysis and morphological evidence, we introduce *Sanguinoderma dehongense* as a new species. In culture, the edges of *S. dehongense* are irregular compared to *S. ovisporum* and *S. concentricum*; on PDA, hyphal growth of *S. dehongense* is denser than that of *S. ovisporum* and *S. concentricum* (Figure 3, Figure 6 and Figure 9). On CMA, *S. dehongense* exhibits the slowest mycelial growth rate compared to *S. ovisporum* and *S. concentricum* (Figure 4, Figure 7 and Figure 10).

**Figure 5 jof-10-00589-f005:**
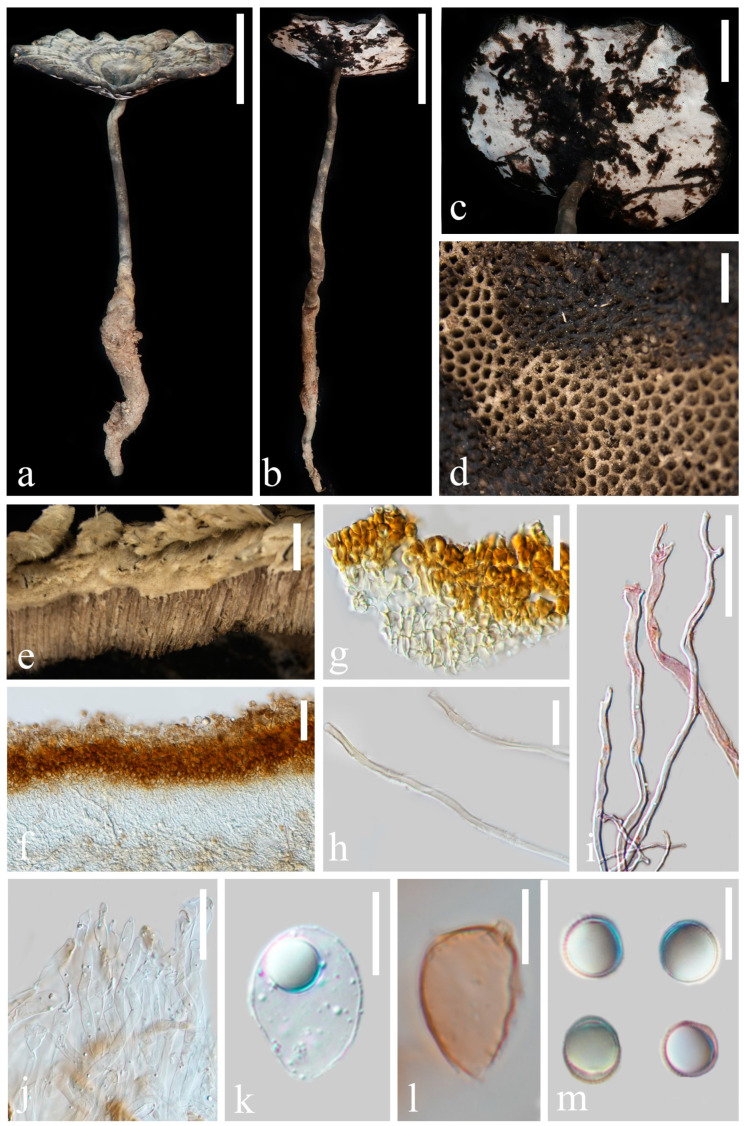
Basidiomata and microscopic structures of *Sanguinoderma dehongense* (HKAS 135636, holotype). (**a**–**c**) Basidiomata. (**d**) Pores. (**e**) Pileus cross-section. (**f**,**g**) Apical cells from pileipellis. (**h**) Skeletal hyphae from context. (**i**) Binding and arboriform skeletal hyphae from context. (**j**) Terminal generative hyphae from tubes. (**k**) Basidioles. (**l**) Basidia. (**m**) Basidiospores. Scale bars: a, b = 5 cm; c = 2 cm; d = 1 mm; e = 2 mm; f, i = 50 μm; g = 20 μm; h, j = 30 μm; k–m = 10 μm.

**Figure 6 jof-10-00589-f006:**
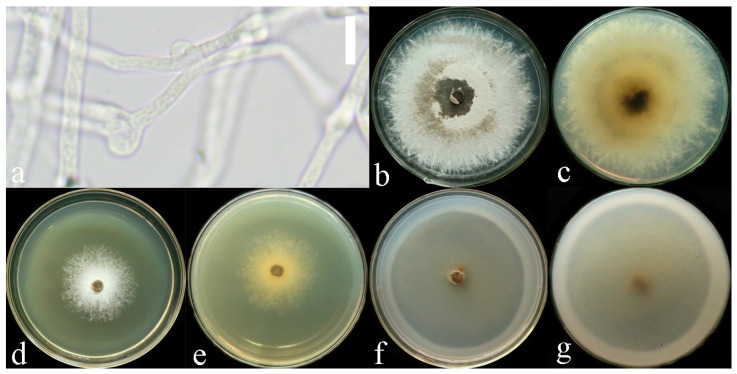
*Sanguinoderma dehongense* culture characters. (**a**) Clamped generative hyphae; (**b**,**c**) colony on PDA; (**b**) obverse; (**c**) reverse; (**d**,**e**) colony on LB; (**d**) obverse; (**e**) reverse; (**f**,**g**) colony on CMA; (**f**) obverse; (**g**) reverse. Scale bars: a = 10 μm.

**Figure 7 jof-10-00589-f007:**
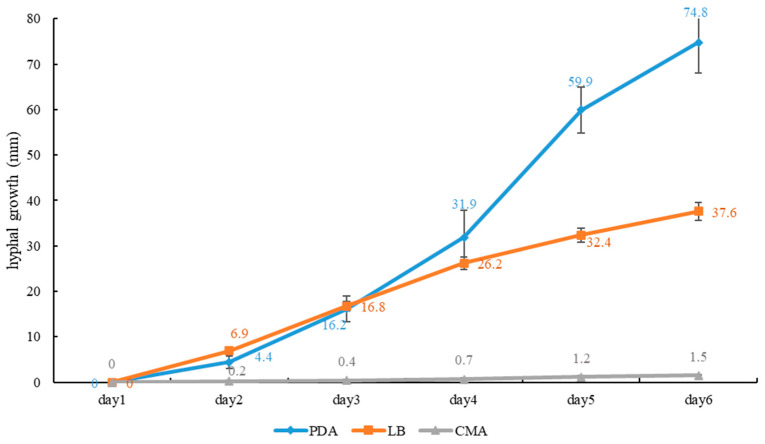
Average daily hyphal growth of *Sanguinoderma dehongense* on different culture media.

***Sanguinoderma ovisporum*** K.Y. Niu, J. He and Z.L. Luo sp. nov. 


**Fungal Names number: FN 572027**


Figure 8 and Figure 11e,f

**Diagnosis.** *Sanguinoderma ovisporum* differs from other species by its dark moderate orange surface pileus, tomentose, lateral stipe, heterogeneous context and ovoid basidiospores (7.5–8.6 × 5.5–7.2 μm).

**Etymology.** The epithet ‘*ovisporum*’ refers to the ovoid basidiospores.

**Holotype.** China, Yunnan Province, Dehong Prefecture, on the ground with humus, 1555 m, Kai-Yang Niu, 16 July 2023, HKAS 135638.

**Description. Basidiomata** is annual, laterally stipitate, solitary, occasionally imbricate and hard corky to hard woody. **Pileus** is single, reniform, up to 11.6 cm in diameter and 4 mm thick. **Pileus surface** is dark moderate orange (#986f53), dull and tomentose with alternating concentric zones mostly desaturated dark orange (#9a765b) and dark moderate orange (#7f553d); dense and radial fine wrinkles. **Pileus margin** is grayish orange (#bab8b3), entire, obtuse to acute and irregularly wavy. **Context** is up to 2 mm thick and homogeneous, the upper layer is very dark grayish yellow (#5f5e5a), the middle layer is dark grayish yellow (#b1b0ad), the lower layer is grayish orange (#b3a696), and it is soft corky and fibrous, without black melanoid lines. **Tubes** are up to 2 mm thick, mostly desaturated dark orange (#917c68), light gray (#d3d3d3) near the pores, hard wood and unstratified. **Pores** are 3–5 per mm, elongated elliptical to subangular and very light gray (#f5f5f5) when fresh, changing to dark red (#9c3d2d) when bruised, then quickly darkening; no discoloration and dissepiments entire when dry. **Stipe** is up to 7 cm long and 2 mm in diameter, cylindrical, hollow, slightly swollen at base, very dark grayish orange (#55423b), darker than pileus and fibrous to woody.

**Hyphal system trimitic**, with generative hyphae 3–6 μm in diameter, hyaline, thin walled, with clamp connections; skeletal hyphae 5–8 μm in diameter, pale yellow, thick walled with a wide or narrow lumen that is subsolid, flexuous and arboriform; binding hyphae are 2–5 μm in diameter, pale yellow, flexuous and branched. All hyphae are IKI− and CB+; tissues darken in KOH. **Pileipellis** is a regular palisade; apical cells are 10–14 × 5–9 μm, short-stick shaped, thin to thick walled and light to dark yellowish brown. **Basidiospores** are subglobose to ellipsoid, faint yellow, IKI− and CB+ with double and slightly thin walls; exospore wall is smooth; endospore wall features indistinct thin pillars, (7.5) 7.7–8.5 (8.6) × (5.5) 5.6–6.9 (7.2) μm, L = 8.2 μm, W = 6.2 μm and Q = 1.31 (40/2). Under SEM, exospore wall is vermiculate, and umbilical protrusion is notably present at the apex (Figure 11e,f). **Basidia** is broadly clavate, hyaline, thin walled and 22–25 × 10–12 μm. **Basidioles** are similar to basidia, hyaline, thin walled and 20–23 × 10–12 μm.

**Culture feature.** Circular, slight smell of corruption and initially white to grayish white; with the increase in culture time, it gradually turns grayish black, changing to blood red when bruised; color does not change with agar-induced growth of the culture; generative hyphae feature multiple branches, with irregularly thickened walls or with scattered thick-walled, refractive areas on walls; texture is sub felty and farinaceous (Figure 3). On PDA, at 24 °C, growth is fast, reaching 65.9 mm in 12 days; culture hyphae are dense. On LB, at 24 °C, growth is slow, reaching 41.3 mm in 12 days; culture hyphae are dense. On CMA, at 24 °C, growth is slow, reaching 35 mm in 12 days; culture hyphae are particularly sparse (Figure 4).

**Additional specimens examined.** China, Yunnan Province, Dehong Prefecture, on the ground with humus, 1322 m, Kai-Yang Niu, 13 July 2023, HKAS 135639.

**Notes:** *Sanguinoderma laceratum* and *S. ovisporum* are similar in having annual, stipitate and soft basidiomata. However, *S. laceratum* is different from *S. ovisporum* in its multiple and superposed pileus, homogeneous context, larger pores (2–3 per mm) and basidiospores (11.3 × 9.7 μm) [8]. In the phylogenetic analyses, *S. ovisporum* was shown to be a distinct lineage in *Sanguinoderma* (Figure 1). In culture, the mycelia of *S. ovisporum*, *S. dehongense* and *S. concentricum* on CMA are all extremely sparse (Figure 3, Figure 6 and Figure 9); the growth rate of *S. ovisporum* was significantly higher than that of *S. dehongense* and *S. concentricum* (Figure 4, Figure 7 and Figure 10).

**Figure 8 jof-10-00589-f008:**
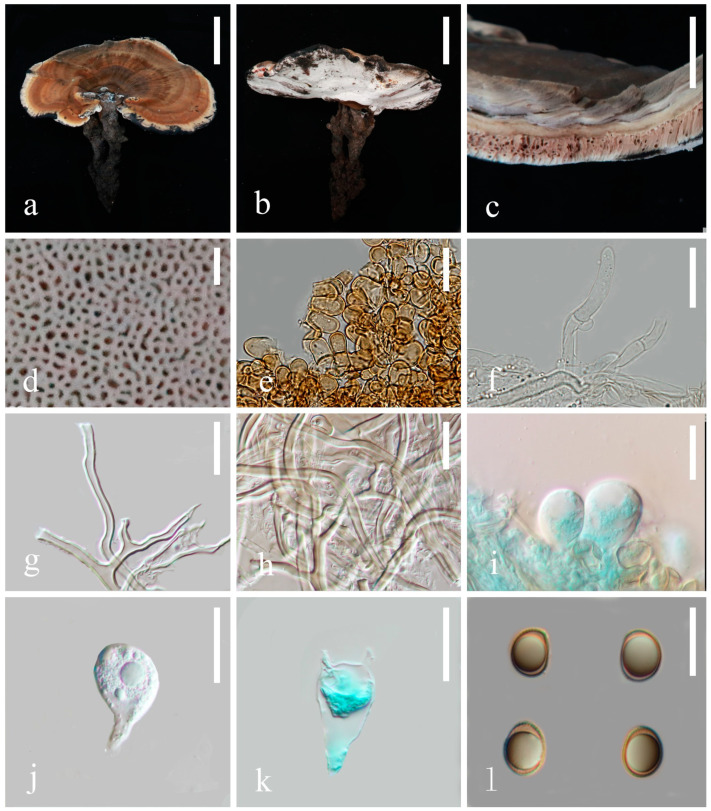
Basidiomata and microscopic structures of *Sanguinoderma ovisporum* (HKAS 135638, holotype) (**a**,**b**) Basidiomata. (**c**) Pileus cross-section. (**d**) Pores. (**e**) Cells from pileus cover. (**f**) Generative hyphae from tubes. (**g**) Skeletal hyphae from context. (**h**) Binding and skeletal hyphae from context. (**i**,**j**) Basidioles. (**k**) Basidia. (**l**) Basidiospores. Scale bars: a, b = 5 cm; c = 5 mm; d = 1 mm; e–h = 20 μm; i–k = 15 μm; l = 10 μm.

**Figure 9 jof-10-00589-f009:**
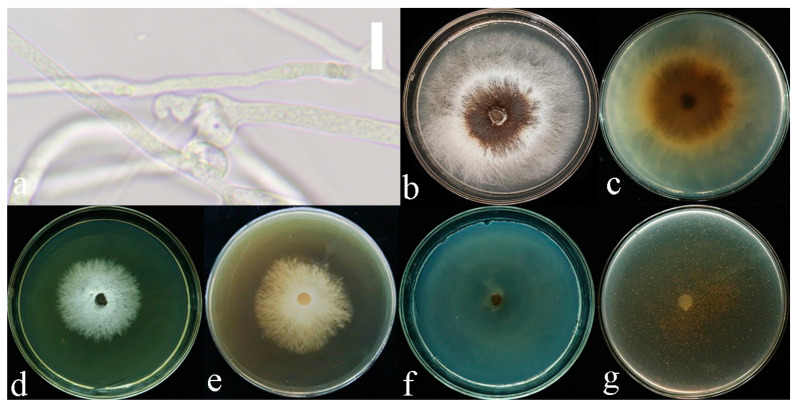
*Sanguinoderma ovisporum* culture characters. (**a**) Clamped generative hyphae; (**b**,**c**) colony on PDA; (**b**) obverse; (**c**) reverse; (**d**,**e**) colony on LB; (**d**) obverse; (**e**) reverse; (**f**,**g**) colony on CMA; (**f**) obverse; (**g**) reverse. Scale bars: a = 10 μm.

**Figure 10 jof-10-00589-f010:**
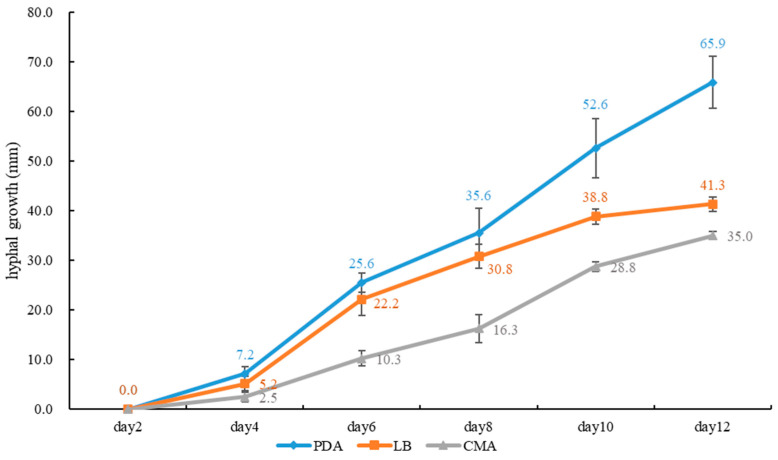
Average daily hyphal growth of *Sanguinoderma ovisporum* on different culture media.

**Figure 11 jof-10-00589-f011:**
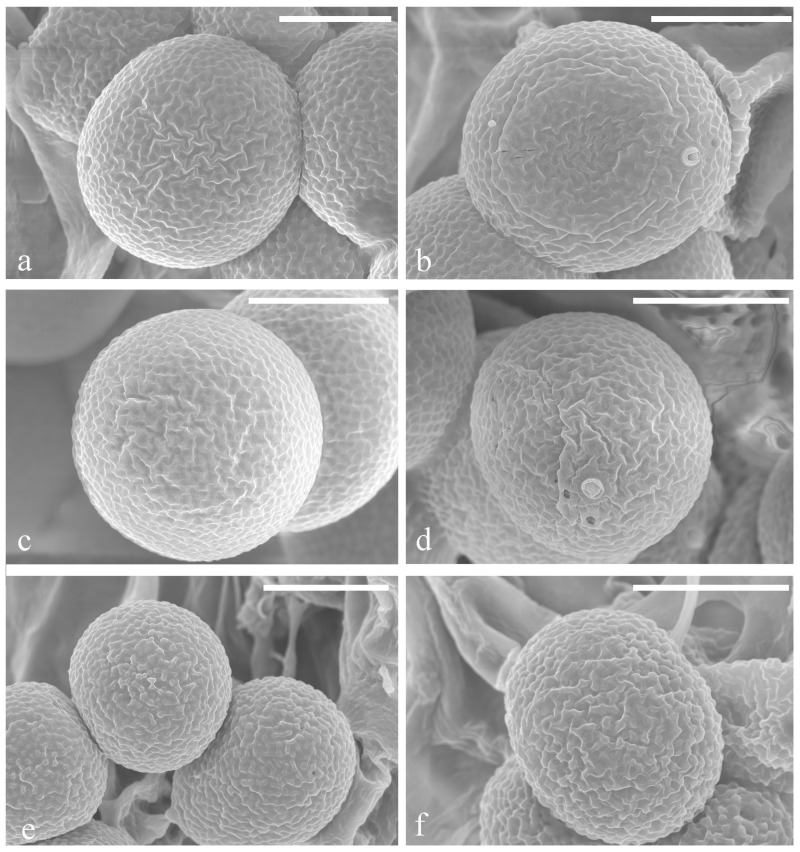
Scanning Electron Micrograph (SEM) of basidiospores of *Sanguinoderma*. (**a**,**b**) *S*. *concentricum* (HKAS 135640, holotype); (**c**,**d**) *S*. *dehongense* (HKAS 135636, holotype); (**e**,**f**) *S*. *ovisporum* (HKAS 135638, holotype). Scale bars: a–f = 5 μm.

## 4. Discussion

*Sanguinoderma* is widely distributed in tropical to subtropical regions [8], there are currently no reports indicating the existence of obligate parasitism and habitats in this genus. It grow in various habitats, such as forest floors, fallen angiosperm trunks and different decaying stumps (*Litchi chinensis*, *Eucalyptus* and *Acacia*) [8,14,20]. For example, *S. rugosum* has been reported on the ground of *Angiosperm* forests and on *Acacia mangium*; more investigation of habitat diversity is an important research direction for the future. So far, a total of 15 species of *Sanguinoderma* have been reported in China. The discovery of three new species of *Sanguinoderma* in this study raises the number of known *Sanguinoderma* species in Yunnan Province to 10, namely *S. ovisporum*, *S. dehongense*, *S. elmerianum*, *S. guangdongense*, *S. laceratum, S. leucomarginatum*, *S. longistipitum*, *S. preussii, S. rugosum* and *S. concentricum*, accounting for 66% of the known species, making it the most diverse province in China. Yunnan Province predominantly experiences tropical and subtropical monsoon climates, as well as rich vegetation types, which provides more possibilities for the growth of *Sanguinoderma* species. Therefore, Yunnan Province may be hiding a greater diversity of *Sanguinoderma* species than expected.

In this study, three new species from Yunnan Province, China are introduced based on morphology and multigene phylogeny. *S. concentricum*, *S. ovisporum* and *S. laceratum* occupied distinct places in the multilocus phylogenetic tree. Morphologically, *S. laceratum* differs from *S. ovisporum* and *S. concentricum* by having lacerate pore dissepiments and fascicular tubes when dry; *S. concentricum* differs from *S. ovisporum* in the pileus margin with lacerated-like petals and a larger basidiospore (9.4–10.9 × 8.1–9.6 μm). *S. dehongense* and *S. elmerianum* formed sister clades; however, the context of *S. elmerianum* features two dark melanoid lines with smaller pores (5–7 per mm) and a shorter stipe (9 cm long) than those of *S. dehongense* [8].

In recent years, phylogenetic analyses have been widely used for the classification of *Ganodermataceae* [8,10,11,14,20,23,36]. Sun et al. [20] conducted phylogenetic analysis of *Ganodermataceae*; single-gene sequences (ITS, nLSU, *rpb*2, mtSSU and nSSU) are difficult to distinguish between some species within *Sanguinoderma*. For instance, a comparison of the ITS, nLSU, *rpb*2, mtSSU and nSSU genes between *S. sinuosum* (MEL 2341763 Type) and *S. rude* (Cui 16592) revealed 1/543 (0.18%, no gaps), 1/875 (0.11%, no gaps), 6/1050 (0.57%, no gaps), 8/513 (1.15%, including one gap), 0/1085 (0%), respectively; these single-gene sequences fail to effectively differentiate the two species. In contrast, *tef*1-α sequences can effectively distinguish species within *Sanguinoderma* in phylogenetic analysis. In this study, a multi-gene approach (ITS, LSU, *rpb*2, *tef*1-α, mtSSU and nSSU) was selected for the phylogenetic analysis of *Sanguinoderma* species to clarify the placement of the newly collected specimens.

In culture, the three strains of *Sanguinoderma* consistently exhibited specific growth characteristics across various media. They displayed the fastest growth rate on PDA, slower growth on LB and the slowest growth on CMA. The PDA medium enriched with glucose likely enhances hyphal growth [37]; so, for isolating *Sanguinoderma* species in culture, prioritizing the use of PDA medium is recommended. All cultures exhibited a color change to brownish red when bruised, progressing to black over time; a characteristic also observed in the basidiomata pores. In addition, in our study, we observed that when using PDA as a preservation medium, cultures stored at 4 °C were prone to viability loss. Therefore, we recommend exploring alternative preservation methods for *Sanguinoderma* strains.

## Figures and Tables

**Table 1 jof-10-00589-t001:** Taxa information and GenBank accession numbers of the sequences used in this study.

Species	Voucher	Locality	ITS	LSU	*rpb*2	*tef*1-α	mtSSU	nSSU	References
*Sanguinoderma bataaense*	Dai 10746	Hainan, China	MK119832	MK119911	MK121511	MK121581	MZ352801	MZ355267	[8,20]
	Cui 6285	Hainan, China	MK119831	MK119910	MK121537	MK121580	MZ352793	MZ355238	[8,20]
	Dai 7862	Hainan, China	KJ531658	-	-	-	-	-	[21]
** *S. concentricum* **	**HKAS 135640 ^T^**	**Yunnan, China**	**PP951682**	**PP951731**	**PP998313**	**PP998318**	**PP988462**	**PP982317**	**This study**
	**HKAS 135641**	**Yunnan, China**	**PP951683**	**PP951732**	**-**	**PP998319**	**PP988463**	**PP982318**	**This study**
** *S. dehongense* **	**HKAS 135636 ^T^**	**Yunnan, China**	**PP947806**	**PP951727**	**PP998309**	**PP998314**	**PP988458**	**PP982313**	**This study**
	**HKAS 135637**	**Yunnan, China**	**PP951679**	**PP951728**	**PP998310**	**PP998315**	**PP988459**	**PP982314**	**This study**
*S. elmerianum*	HMAS 133187	Yunnan, China	MK119834	MK119913	-	-	MZ352824	MZ355234	[8,20]
	Dai 20634	Yunnan, China	MZ354875	MZ355082	-	MZ221724	MZ352821	MZ355148	[8,20]
	Cui 8940	Guangdong, China	MK119833	MK119912	-	-	MZ352812	MZ355305	[8,20]
*S. flavovirens*	Cui 16935 ^T^	Zambia	-	MK119914	MK121532	MK121582	MZ352811	MZ355254	[8,20]
*S. guangdongense*	Cui 17259 ^T^	Guangdong, China	MZ354877	MZ355123	MZ358834	MZ221726	MZ352816	MZ355139	[20]
	Dai 16724	Thailand, China	MZ354876	MZ355117	MZ358833	MZ221725	MZ352815	MZ355271	[20]
	Dai 20419	Yunnan, China	MZ354890	MZ355083	MZ358835	MZ221727	MZ352818	MZ355155	[20]
*S. infundibulare*	Dai 18149 ^T^	Guangdong, China	MK119847	MK119926	MK121529	MK121597	MZ352790	MZ355239	[8,20]
	URM 450213	Ecuador	MK119849	MK119927	-	-	MZ352792	MZ355252	[8,20]
	Cui 17238	Guangdong, China	OM780277	-	MZ358837	MZ221729	MZ352800	MZ355149	[20]
*S. laceratum*	Cui 8155 ^T^	Yunnan, China	NR174040	MK119928	-	-	MZ352810	-	[8,20]
*S. leucomarginatum*	Dai 12264	Yunnan, China	OP700311	OP700344	OP696845	OP696857	OP703259	OP700325	[14,20]
	Dai 12377 ^T^	Yunnan, China	OP700312	OP700345	OP696846	OP696860	OP703260	OP700326	[14]
	Dai 12362	Yunnan, China	KU219986	KU220009	OP696847	OP696858	OP703261	OP700327	[22]
*S. longistipitum*	Dai 20696 ^T^	Yunnan, China	MZ354881	MZ355084	-	MZ221732	MZ352822	MZ355145	[20]
	Cui 13903	Hainan, China	MZ354882	MZ355114	MZ358839	MZ221733	MZ352809	MZ355301	[20]
	Dai 16635	Thailand, China	MZ354883	MZ355120	MZ358840	MZ221734	MZ352802	MZ355260	[20]
*S. melanocarpum*	Dai 18512	Malaysia	MZ354888	MZ355118	-	MZ221735	MZ352794	MZ355313	[20]
	Dai 18603 ^T^	Malaysia	MZ354889	MZ355113	MZ358841	MZ221736	MZ352796	MZ355281	[20]
*S. microporum*	Cui 13851 ^T^	Hainan, China	MK119854	MK119933	MK121512	MK121602	MZ352797	MZ355270	[8,20]
	Cui 14022	Guangxi, China	MK119856	MK119935	MK121515	MK121604	MZ352798	MZ355298	[8,20]
	Cui 16335	Guangxi, China	MK119857	MK119936	MK121514	MK121605	OP703262	OP700328	[14,20]
	Cui 14001	Guangxi, China	MK119855	MK119934	MK121513	MK121603	OP703263	OP700329	[14,20]
*S. microsporum*	Dai 16726 ^T^	Thailand, China	-	MZ355119	-	MZ221737	MZ352795	MZ355272	[20]
	Cui 13897	Hainan, China	MZ354878	MZ355127	-	MZ221739	MZ352804	MZ355300	[20]
	Cui 13901	Hainan, China	MZ354879	MZ355121	-	MZ221738	MZ352803	MZ355299	[20]
** *S. ovisporum* **	**HKAS 135638 ^T^**	**Yunnan, China**	**PP951680**	**PP951729**	**PP998311**	**PP998316**	**PP988460**	**PP982315**	**This study**
	**HKAS 135639**	**Yunnan, China**	**PP951681**	**PP951730**	**PP998312**	**PP998317**	**PP988461**	**PP982316**	**This study**
*S. perplexum*	Cui 6496	Hainan, China	KJ531650	KU220001	MK121538	MK121583	MZ352825	MZ355263	[20,21]
	Cui 6554	Hainan, China	MK119835	MK119915	MK121540	MK121585	MZ352826	MZ355264	[8,20]
	Dai 10811	Hainan, China	KJ531651	KU220002	MK121539	MK121584	MZ352827	MZ355302	[20,21]
	Wei 5562	Hainan, China	KJ531652	-	-	-	-	-	[21]
*S. preussii*	Dai 20438	Yunnan, China	OP700314	OP700347	OP696848	OP696869	OP703265	OP700331	[14]
	Dai 20622	Yunnan, China	OP700315	OP700348	-	OP696862	OP703266	OP700332	[14]
	Dai 20624	Yunnan, China	OP700316	OP700349	-	OP696863	OP703267	OP700333	[14]
*S. reniforme*	Cui 16511 ^T^	Zambia	NR174041	MK119929	MK121531	MK121599	-	MZ355322	[8,20]
*S. rude*	MEL 2317411	Australia	MK119842	-	MK121524	MK121592	MZ352819	MZ355306	[8,20]
	DHCR457	Brazil	MN077517	MN077551	-	MN061693	-	-	[23]
	Cui 16592	Australia	MK119836	MK119916	MK121521	MK121586	MZ352924	MZ355307	[8,20]
*S. rugosum*	Cui 17260	Guangdong, China	OP700317	OP700350	OP696849	OP696859	OP703270	OP700336	[14]
	Cui 14033	Guangxi, China	OP700318	OP700351	OP696850	OP696864	OP703271	OP700337	[14]
*S. sinuosum*	MEL 2341763 ^T^	Australia	MK119853	MK119931	MK121525	MK121601	MZ352820	MZ355291	[8,20]
	MEL 2366586	Australia	MK119852	MK119930	MK121527	MK121600	MZ352920	MZ355261	[8,20]
*S. tricolor*	Cui 18242	Malaysia	MZ354992	MZ355099	MZ358843	MZ221743	MZ352829	MZ355303	[20]
	Cui 18292 ^T^	Malaysia	-	MZ355101	-	MZ221742	MZ352828	MZ355273	[20]
	Dai 18574	Malaysia	MZ354993	MZ355102	MZ358844	MZ221744	MZ352830	MZ355265	[20]
*Magoderna subresinosum*	Dai 18626	Malaysia	MK119823	MK119902	MK121507	MK121571	MZ352831	MZ355211	[8,20]
	Cui 18262	Malaysia	MZ354871	MZ355088	-	-	MZ352832	MZ355258	[20]

New sequences generated are shown in bold. ^T^ Indicates the type specimens.

## Data Availability

Not applicable.

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
