# Peer review of "Morphological and Phylogenetic Analyses Reveal Three Novel Species of Sanguinoderma (Ganodermataceae, Basidiomycota) from Yunnan Province, China"

_jof, 2024, doi:10.3390/jof10080589_

Round 1

Reviewer 1 Report

Dear authors,

I believe that you planned the experiment very well, so that in addition to complex morphological tests, you also correctly applied phylogenetic analysis using a multi-gene approach. With this, you managed to characterize the newly discovered species of mushrooms in detail and define their place in the phylogenetic tree with certainty.

I congratulate you on an excellently conducted study and conclusions and I have no further questions or concerns

Author Response

Thank you for your comments. 

Wish you all the best!

Kai-Yang Niu

Reviewer 2 Report

In general the work has quality enough and the information is detailed. However it is missing some information, specially within methods. And the Discussion looks poor to my opinion, it should be rewrite/improve or avoid and change to conclusions. More detailed comments with the aim to improve and clarify the work are within detalied comments.

Find my comments attached in a word file.

Author Response

Thank you for your feedback. In this revision, substantial adjustments and modifications have been made to the methodology and discussion sections.

Wish you all the best!

Kai-Yang Niu

Reviewer 3 Report

This is an interesting and useful manuscript describing three new species of Sanguinoderma; however, it needs much work before it can be published. Some of the comments are indicated below and many directly in the manuscript, but I am sure that many corrections still need to be made.

The manuscript needs a thorough linguistic revision, as it uses both American and British English and has many inconsistencies and problems. I am not a native English speaker, so I cannot notice all the errors. I made some suggestions but still need many more changes.

In some parts the descriptions of the species are very similar to those in Sun et al. (2020, 2022), so a plagiarism test would detect a high percentage of similarity. I changed some parts but there is still work to be done.

Phylogenetic results should be presented as a separated part of the taxonomic descriptions of the new species.

Cutural descriptions must be improved and based in some literature referring to culture descriptions, as Nobles (1965. Identification of cultures of wood-inhabiting Hymenomycetes. Can. Jour. Bot. 43: 1097–1139).

Many corrections are indicated only once but must be corrected throughout the manuscript.

Review that the “Diagnosis” section includes, to the extent possible, unique characteristics of the species in question.

Please check the comments in the attached file.

Author Response

Thank you very much for your valuable feedback. I will address the major comments you raised below. The additional details of the revisions are provided in the attachment.

  1. The manuscript needs a thorough linguistic revision, as it uses both American and British English and has many inconsistencies and problems. I am not a native English speaker, so I cannot notice all the errors. I made some suggestions but still need many more changes.

A: Thank you for your assistance and revisions. We have made numerous improvements and modifications to the existing content.

  1. In some parts the descriptions of the species are very similar to those in Sun et al. (2020, 2022), so a plagiarism test would detect a high percentage of similarity. I changed some parts but there is still work to be done.

A: Under your suggestions, we have completed revisions to all relevant content, and some parts differ slightly from those in Sun et al.

  1. Phylogenetic results should be presented as a separated part of the taxonomic descriptions of the new species.

A: We have proposed to present the phylogenetic results as a separate section.

  1. Cutural descriptions must be improved and based in some literature referring to culture descriptions, as Nobles (1965. Identification of cultures of wood-inhabiting Hymenomycetes. Can. Jour. Bot. 43: 1097–1139).

A: We have revised the description of Cutural based on the references you provided.

  1. Review that the “Diagnosis” section includes, to the extent possible, unique characteristics of the species in question.

A: The "Diagnosis" section has been further refined.

Wish you all the best!

Kai-Yang Niu

Reviewer 4 Report

The manuscript "Morphological and phylogenetic analyses reveal three novel species of Sanguinoderma (Ganodermataceae, Basidiomycota) from Yunnan Province, China," is well-researched and provides valuable insights into these newly discovered species. However, a few areas need further attention to enhance the manuscript. 

The discussion on the phylogenetic analysis should be expanded to explain better how these new species fit into the Sanguinoderma genus and how they relate to other similar fungi. 

The manuscript would be improved by adding more phylogenetic analysis and trying different methods to confirm the findings. Adding extra figures to show these analyses would help support the conclusions and strengthen the study.

The manuscript "Morphological and phylogenetic analyses reveal three novel species of Sanguinoderma (Ganodermataceae, Basidiomycota) from Yunnan Province, China," is well-researched and provides valuable insights into these newly discovered species. However, a few areas need further attention to enhance the manuscript. 

The discussion on the phylogenetic analysis should be expanded to explain better how these new species fit into the Sanguinoderma genus and how they relate to other similar fungi. 

The manuscript would be improved by adding more phylogenetic analysis and trying different methods to confirm the findings. Adding extra figures to show these analyses would help support the conclusions and strengthen the study.

Minor comments 

1. Please include the specific version numbers of the software tools used for your analyses (e.g., MAFFT, BioEdit, RAxML, MrBayes). This information is crucial for reproducibility.

2. For any online tools used, please provide the website URLs and the dates when you accessed them.

3. Please include MycoBank numbers for all newly described species.

Author Response

Thank you very much for your valuable feedback. I will address the issues you raised below.

  1. The discussion on the phylogenetic analysis should be expanded to explain better how these new species fit into the Sanguinoderma genus and how they relate to other similar fungi.

A: Thank you for your suggestions. We have made extensive revisions and improvements to the discussion section.

  1. Please include the specific version numbers of the software tools used for your analyses (e.g., MAFFT, BioEdit, RAxML, MrBayes). This information is crucial for reproducibility.

A: Thank you for your suggestions. We have added the corresponding software version numbers.

  1. For any online tools used, please provide the website URLs and the dates when you accessed them.

A: Thank you for your suggestions. We have completed the addition of the corresponding content.

  1. Please include MycoBank numbers for all newly described species.

A: In the article, we provided the Fungal Names number, which can also serve as the number for the new species, similar to the MycoBank numbers.

  1. The manuscript would be improved by adding more phylogenetic analysis and trying different methods to confirm the findings. Adding extra figures to show these analyses would help support the conclusions and strengthen the study.

A: In this article, the phylogenetic analysis employed Maximum Likelihood Analysis and Bayesian Analysis, incorporating all sequences of the genus. The corresponding data robustly supports the current argument.

Wish you all the best!

Kai-Yang Niu